# Changes in Inequality in Use of Maternal Health Care Services: Evidence from Skilled Birth Attendance in Mauritania for the Period 2007–2015

**DOI:** 10.3390/ijerph19063566

**Published:** 2022-03-17

**Authors:** Mohamed Vadel Taleb El Hassen, Juan M. Cabases, Moulay Driss Zine Eddine El Idrissi, Samuel Mills

**Affiliations:** 1World Bank Group, Washington, DC 20433, USA; mzineeddineelidr@worldbank.org (M.D.Z.E.E.I.); smills@worldbank.org (S.M.); 2Department of Economics, Public University of Navarra, 31006 Pamplona, Spain; jmcabases@unavarra.es

**Keywords:** skilled birth attendance, inequality, concentration curve, concentration index, oaxaca decomposition, Mauritania

## Abstract

Skilled birth attendance is critical to reduce infant and maternal mortality. Health development plans and strategies, especially in developing countries, consider equity in access to maternal health care services as a priority. This study aimed to measure and analyze the inequality in the use of skilled birth attendance services in Mauritania. The study identifies the inequality determinants and explores its changes over the period 2007–2015. The concentration curve, concentration index, decomposition of the concentration index, and Oaxaca-type decomposition technique were performed to measure socioeconomically-based inequalities in skilled birth attendance services utilization, and to identify the contribution of different determinants to such inequality as well as the changes in inequality overtime using data from Mauritania Multiple Indicator Cluster Surveys (MICS) 2007 and 2015. The concentration index for skilled birth attendance services use dropped from 0.6324 (*p* < 0.001) in 2007 to 0.5852 (*p* < 0.001) in 2015. Prenatal care, household wealth level, and rural−urban residence contributed most to socioeconomic inequality. The concentration index decomposition and the Oaxaca-type decomposition revealed that changes in prenatal care and rural−urban residence contributed positively to lower inequality, but household economic status had an opposite contribution. Clearly, the pro-rich inequality in skilled birth attendance is high in Mauritania, despite a slight decrease during the study period. Policy actions on eliminating geographical and socioeconomic inequalities should target increased access to skilled birth attendance. Multisectoral policy action is needed to improve social determinants of health and to remove health system bottlenecks. This will include the socioeconomic empowerment of women and girls, while enhancing the availability and affordability of reproductive and maternal health commodities. This policy action can be achieved through improving the availability of obstetric service providers in rural areas; ensuring better distribution and quality of health infrastructure, particularly health posts and health centers; and, ensuring user fees removal for equitable, efficient, and sustainable financial protection in line with the universal health coverage objectives.

## 1. Introduction

Economic performance in Mauritania has improved substantially over the past 15 years. The country underwent rapid economic growth from 2008 to 2014, accompanied by significant progress in household welfare. This was reflected by a sharp decline in poverty rate, affecting 44.5% of the country’s population in 2008, but falling to 33% in 2014 [1,2]. These improvements are explained by recoveries in production, productivity, and income in rural areas [3]. However, Mauritania’s health system continues to perform poorly on indicators of maternal and child health care services. The maternal mortality rate (MMR), with 582 deaths per 100,000 live births in 2015, was not even close to the Millennium Development Goals (MDG) target of 232 deaths per 100,000 live births [4,5], and is also among the highest in the region, albeit near the sub-Saharan Africa average of 542 [6].

The challenge for the Mauritanian health sector has been to maximize access to health services, despite the constraints that inevitably accompany a very low population density, estimated at 4.3 inhabitants per km^2^, compared to an average of 44.9 per km^2^ in sub-Saharan Africa [7]. The key strategy to improve access is extending the coverage of the basic infrastructure to bring 91% of the population to access within 5km of a health post or a functional health center, providing access to the three most common essential health service packages [8]. However, geographic access to health services within a 5 km radius of a health facility is stands at 67.3%, with disparity among regions [2]. The regions with a high incidence of poverty are those with poor geographic access to health services. Poor road conditions result in high transport costs and long travel times, especially during obstetric emergencies, such as deliveries. In urban centers such as the capital, Nouakchott, the problem of geographic accessibility to health services exists, but to a much lesser extent, and the private sector has helped to alleviate this problem. The Living Standards Measurement study undertaken in 2014 [2] shows that 15.7% of the Nouakchott population resides more than 5 km from the nearest facility. 

The availability of basic reproductive, maternal, neonatal and child health (RMNCH) services remains low, aggravated by a lack of infrastructure and health personnel, and often falling below 60% [9]. For example, 41% of public health facilities and 80% of private health facilities do not provide assisted delivery services to the population nominally covered [9]. 

As in other sub-Saharan countries, numerous strategies for reducing financial barriers have been attempted. These include community-based health insurance (CBHI), an indigence fund that aims at exempting the poor from paying for health care at public facilities, an obstetric risk insurance scheme, and compulsory health insurance through the national health insurance fund. Furthermore, private insurance companies are generally contracted by large enterprises for the benefit of their employees, but their coverage remains limited to the affluent segments of the population. The obstetric risk insurance scheme is based on a set of basic gynecological and obstetric services provided by health facilities against a single contribution, roughly equivalent to USD$20, covering prenatal care, delivery including caesarean section, and treatment of obstetric complications. Despite these efforts, significant financial barriers persist. The community-based health insurance is limited to a few experiences in two districts with extremely low coverage. Waelkens Maria-Pia et al. (2017) [10] found that the membership of the CBHI remained far too low to be regarded as a potential stepping stone on the path towards universal health coverage in Mauritania. Philibert et al. (2017) [11] also found that the obstetric risk insurance scheme had not given pregnant women equitable access to facility-based health care, instead it appeared to show a bias in favor of certain social strata. In addition, the national health insurance fund, barely covered 12% of the population in 2019 [12]. The population covered under the national health insurance fund is composed of civil servants and formal private sector workers. Thus, households continue to spend relatively large sums on of out-of-pocket health care expenditures, amounting to 51% of the total health expenditure in 2016 [13]. 

Equity in access and use of maternal health care services is considered a priority by most national health development plans and strategies in sub-Saharan Africa countries, which have committed to reducing the maternal mortality ratio to 70 per 100,000 live births by 2030, in line with the Sustainable Development Goals [14]. In order to reduce perinatal, neonatal, and maternal deaths, the National Strategy for Accelerated Growth and Shared Prosperity (2016–2030), as well as the National Health Development Plan (2012–2020), both prioritize access to pivotal high-quality maternal health interventions, including skilled birth attendance (SBA), a priority to reduce perinatal, neonatal, and maternal deaths [15,16]. SBA occurs when a midwife, physician, obstetrician, or nurse provides essential and emergency health care services to women and their newborns during pregnancy, childbirth, and the postpartum period [17]. Progress in raising the coverage of births attended by skilled health personnel has been slow since 2007: increasing from 60.9% in 2007 to only 69.3% in 2015, reflecting lack of universal access to maternal care [15].

The present study contributes to the literature that examines the trends and assesses the magnitude of inequalities in the use of skilled birth attendance. Previous studies conducted in Kenya, Nigeria, Bangladesh, Namibia [18,19,20,21], and other developing countries [22,23,24,25,26,27,28,29,30,31,32] have indicated that socioeconomic status, mother’s level of education, urban−rural residence, birth order, and insurance status play a key role in explaining the differences in use and the levels of inequality in skilled birth attendance and maternal health care uptake. A few studies have decomposed income inequalities and measured change in inequality in maternal health care use in sub-Saharan Africa [33,34], while, to the best of our knowledge, no such studies have been conducted in Mauritania.

The present study seeks to quantify the socioeconomic trends behind inequality in skilled birth attendance in Mauritania, and measures the contribution of the determinants of inequality, by means of the concentration curve, concentration index, decomposition of the concentration index, and Oaxaca-type decomposition technique. The study uses the most recent data (2015) and the earliest available (2007). The 2015 data coincides with the end of the 15-year run-up to the Millennium Development Goals 2015 and the mid-term review of the Mauritanian National Health Development Plan (2012–2020). The results of our analysis should help to facilitate the evaluation of the strategies developed by the Mauritanian government to promote equitable access to maternal health care, and also to inform public policies aimed at effectively reducing the MMR.

## 2. Materials and Methods

### 2.1. Data Sources and Sample

The present study was based on publicly available data from the 2007 and 2015 multiple indicator cluster surveys (MICS). The MICS are nationally representative surveys of households, women of childbearing age, children under five, and men aged 15 to 49, gathering information about reproductive, maternal, neonatal, and child health care. The surveys also cover early childhood development, education, and literacy. The survey results are used to monitor and evaluate a range of indicators in the areas of reproductive, maternal, neonatal, and child health; education; child protection; and HIV/AIDS. These indicators are monitored by the National Strategy for Accelerated Growth and Shared Prosperity (2016–2030) and the National Health Development Plan (2012–2020). The surveys are typically conducted and implemented by the National Statistical Office at five-year intervals [35,36].

In the 2007 survey, the sample consisted of 11,000 households based on the 2000 population census, while in 2015, the sample consisted of 11,985 households based on the 2013 population census. The sample included in the MICS 2007 was obtained by using a two-stage stratified cluster sampling approach, whereas the 2015 sample was derived from a three-stage stratified cluster sampling approach. In 2007, interviews were conducted with a total of 12,549 women aged 15–49 years from the selected households, compared with 14,342 women from the same age bracket in 2015 [35].

The overall sample sizes of the two rounds differs slightly because the two surveys used two different censuses of the population, taken 2000 and 2013, respectively.

The final sample of the study used a total of 3539 women aged 15–49 years old in 2007 and 4172 women aged 15–49 years old in 2015, who had given birth in the previous two years.

### 2.2. Measuring and Decomposing Socio-Economic Inequalities in SBA

#### 2.2.1. Measuring Inequality 

To measure inequality, we used the concentration curve (CC) and concentration index as a conventional method for computing and decomposing socioeconomic related inequality in skilled birth attendance. The CC plots shares of the SBA against quintiles of wealth index. The concentration index (CI) is directly related to the concentration curve (CC) and represents twice the area between the CC and the 45-degree or equality line [36,37]. The standard concentration index (CI) can be computed as follows [38]: (1)$$CI=\frac{2}{\bar{y}}COV\left(y_{i};r_{i}\right)$$
where y is the variable of interest of the *i* individual, $\bar{y}$ is the mean of $y_{i}$, and $r_{i}$ is the fractional rank of the ith person. The value of CI can range from −1 to +1. A positive value suggests a pro-rich concentration of skilled birth attendance and a negative value indicates a pro-poor distribution. When *CI* takes a positive value, CC would be presented below the line of equality (45-degree line), while if the CI takes a negative value, CC will lie above the line of equality. If the value of CI is zero, this indicates that there is no income related inequality in the distribution of y and the CC will coincide with the line of equality.

When y is a binary variable, CI ranges from $\bar{y}-1$ to $1-\bar{y}$. To allow for comparisons between different time periods, it has to be corrected in some way [39,40,41,42,43]. Erreygers G. proposes the following correction:(2)$$CIE=\frac{4\bar{y}}{y^{max}-y^{min}}CI\;$$
where *CIE* is the corrected concentration index.

#### 2.2.2. Decomposing Inequality 

When there is a linear relationship between health outcome, skilled birth attendance in our case ($y_{i}$) and a set of *k* explanatory variables $x_{ki}$ [44,45,46,47], SBA can be specified using the following multivariate linear regression equation: (3)$$y_{i}=\alpha +\sum_{k}\beta_{k}{\bar{x}}_{ki}+\varepsilon_{i}$$
where

$y_{i}$ is skilled birth attendance ($y_{i}\;$= 1 if birth was attended by a skilled health professional and $y_{i}$= 0 if not)$x_{ki}$ is a set of k explanatory variables for skilled birth attendance;$\beta_{k}$ is the regression coefficient of the explanatory variables $x_{ki}$;$\varepsilon_{i}$ is the error term.

Then *CIE* can be expressed as weighted sum of partial concentration indices for the explanatory variables of inequality, being the weight of the elasticity of γ with respect to $x_{k}$ [44,45,46,47]:(4)$$CIE=\sum_{k}\left(\frac{\beta_{k}{\bar{x}}_{k}}{\bar{y}}\right)CIE_{k}+\frac{GCIE_{\varepsilon }}{\bar{y}}$$
where $CIE_{k}$ is the concentration index for $x_{k}$, $\bar{x}$ is the mean of $x_{k}$, $\bar{y}$ is the mean of SBA (*y*), and $GCIE_{\varepsilon }$ is the generalized concentration index for the term (ε). 

Considering the dichotomous characters of SBA, a non-linear model is required. We used probit models to carry out all the estimates, then a linear approximation was performed for the decomposition analysis based on Erreyger’s normalized concentration index [42,43], as follows: (5)$$CIE=4\;\sum_{k}\left(\beta_{k}^{m}{\bar{x}}_{k}\right)CIE_{k}+GCIE_{\varepsilon }$$
where $\beta_{k}^{m}$ represents the partial effects ($\frac{dy}{dx_{k}}$) evaluated at the sample means.

#### 2.2.3. Decomposing Changes in Inequality

The change of *CIE* over the period 2007–2015 can be expressed as follow:(6)$$△CIE_{y}=CIE_{y2015}-CIE_{y2007}$$

To examine the drivers of changes in equality in skilled birth attendance in 2007 and 2015, we applied the Oaxaca decomposition to Equation (5). This leads to the following
(7)$$△CIE=\sum_{k}\eta_{k2015}\left(CIE_{k2015}-CIE_{k2007}\right)+\sum_{k}CIE_{k2007}\left(\eta_{k2015}-\eta_{k2007}\right)+△\left(\frac{GCIE_{\varepsilon t}}{{\bar{y}}_{t}}\right)$$
or alternatively:(8)$$△CIE=\sum_{k}\eta_{k2007}\left(CIE_{k2015}-CIE_{k2007}\right)+\sum_{k}CIE_{k2015}\left(\eta_{k2015}-\eta_{k2007}\right)+△\left(\frac{GCIE_{\varepsilon t}}{{\bar{y}}_{t}}\right)$$
where $\eta_{kt}$ =$\;\left(\beta_{k}^{m}{\bar{x}}_{k}\right)$ is the elasticity of y with respect to kth determinant in year t. 

Aside from the contribution of unexplained factors $△\left(\frac{GCIE_{\varepsilon t}}{{\bar{y}}_{t}}\right)$, the changes of $CIE_{y}$ can be explained by changes in elasticities $(\eta_{k2015}-\eta_{k2007}$) and by changes in the concentration index $\left(CIE_{k2015}-CIE_{k2007}\right)$, or both. The elasticity of the kth determinant $\eta_{kt}$ =$\;\left(\beta_{k}^{m}{\bar{x}}_{k}\right)$, can change due to variations in any of its components, namely $\beta_{k}^{m}$ and ${\bar{x}}_{k}$. 

### 2.3. Definition of Variables

#### 2.3.1. Outcome Variable

Our variable of interest is birth attendance by a qualified health professional. The skilled birth attendant is “an accredited health professional—such as a midwife, doctor, or nurse—who has been educated and trained to proficiency in the skills needed to manage normal (uncomplicated) pregnancies, childbirth, and the immediate postnatal period, and in the identification, management, and referral of complications in women and newborns” [17]. Among the respondents who declared having given birth in the two years preceding the survey, the variable of interest was elicited by means of the question “Who assisted you during the delivery?”. The possible response options were doctor, nurse, midwife, auxiliary birth attendant, traditional birth attendant, community health worker, parent/friend, others, or nobody. Based on the World Health Organization (WHO) definition, the outcome variable was dichotomized to 1 if the birth was attended by a doctor, nurse, or midwife, and 0 otherwise. 

The following explanatory variables were selected based on previous studies [18,19,20,21,22,23,24,25,26,27,28,29,30,31,32] and their availability in the dataset.

#### 2.3.2. Socioeconomic Indicator

MICS surveys lack information about household income or consumption as a measure of living standards. Therefore, household wealth index (WI) was built using principal component analysis (PCA). The variables included in the PCA were the availability of water and sanitation, radio, television, satellite dish, landline telephone, mobile phone, refrigerator, oven stove, air conditioner, fan, washing machine, electricity generator, solar panel, watch, bicycle, motorcycle or scooter, cart, car/truck, motor boat, computer, and internet connections. Each household in the total sample was then assigned a wealth score based on its assets held and on the final factor scores obtained. We used a household wealth index (WI) provided in MICS to depict socioeconomic status and living standards. For calculation of the concentration index, WI was used as a continuous variable, and for the decomposition analysis, WI was categorized into quintiles with the first quintile representing the poorest and the fifth quintile representing richest.

#### 2.3.3. Social Determinants of SBA

Socio-demographic characteristics were proxied by women’s age (15–24, 25–34, 35–44, or 45–49 years old), four or more offspring (because that may affect not only the health, but also the wealth status of women), prenatal care as a potential determinant associated with skilled birth attendance, maternal level of education (no education, primary, secondary, or higher), place of residence (urban or rural), 13 regions of the country (Nouakchott, Hodh Charghy, and so forth), and wealth quintile. Individual weights were applied to the data to make the sample representative of the whole population. All analyses were conducted in StataSE-15©. The concentration indices were calculated by using the conindex command in Stata [48].

## 3. Results

### 3.1. Descriptive Statistics

Most of the study respondents were 15–24 years old, rural residents, and from the Nouakchott region both in 2007 and 2015 (Table 1). About 57% of the deliveries were attended by skilled birth attendants in 2007 compared with 64% in 2015. A breakdown by various maternal and household characteristics displays the associated changes in SBA. The most pronounced differences are seen in association with household wealth quintiles, urban rural residency, and women’s level of education. Skilled birth attendance in the richest quintile is about 70% higher in both surveys than in the poorest quintile. The average of skilled birth attendance among educated women is almost twice that of women with no education in both surveys (Table 1). Furthermore, the weighted means of SBA in urban areas is twice that of rural areas.

In both surveys, the use of skilled birth attendants is over 80% in the Nouakchott, Nouadibou, Trarza, Tiris-Zemour, and Inchiri regions, while the lowest (about 25–50%) is seen in Guidimagha, Hodh Charghy, Hodh Gharby, and Gorgol. In 2015, parturient women in Nouakchott, Tiris-Zemour, and Nouadibou used the services of skilled attendants by more than 64% compared to those living in Guidimagha. In 2015, the use of skilled birth attendance was below the national average of 64.41% in six of the 13 regions.

### 3.2. Measuring Inequality

The concentration curves for inequalities of SBA in 2007 and 2015 are shown in Figure 1. Concentration curves of skilled birth attendance use for both surveys lie below the line of equality, indicating that skilled birth attendance care is more frequently used among richer women. Moreover, the corrected concentration indices were positive 0.6324 (*p* < 0.001) in 2007 and 0.5852 (*p* < 0.001) in 2015, indicating that the better-off women are significantly more likely to be attended by skilled professionals at the time of delivery than their poorer counterparts. The decrease in the concentration index in 2015 indicates a slight decrease in the inequalities in skilled birth attendance between 2007 and 2015.

### 3.3. Decomposition of SBA Inequalities

The results of the decomposition of SBA inequalities in 2007 and 2015 are reported in Table 2. The table shows the marginal effects, the elasticity, and the corrected concentration indices of the regressors (*CIE*), as well as the absolute and percentage of contributions of explanatory variables and their changes. The marginal effect estimates in both surveys (2007 and 2015) show that living in a rural area; living in Hodh Charghy, Hodh Gharby, Assaba, Gorgol, or Guidimagha; and being poor (Q2) or the poorest (Q1) were associated with lower use of skilled birth attendance compared to living in urban areas, living in Nouakchott, or being rich (Q5). In contrast, prenatal care use and being aged of 35–49 were associated with high use of SBA compared to non-utilization of prenatal care during the pregnancy or being in the 15–24 age group. In addition, there are some significant changes in the partial effects in both surveys. In particular, living in Tagant and having received primary education were associated with lower use of SBA in 2015 compared to 2007. Conversely, having one or more living children was associated with high use of SBA in 2007 compared to 2015. Moreover, regions showed different effects from one period to another: in nine of them, it was observed a noticeable decrease in the partial effect, whereas a marked increase was registered in three regions, namely Assaba, Brakna, and Tagant.

The elasticity reveals the sensitivity of SBA to a change in the determinants. Any variable with a positive elasticity suggests that women with this characteristic are more likely to receive skilled birth attendance at delivery. The concentration indices were estimated for each regressor.

As explained in the previous section, corrected concentration indices decreased from 0.6324 (*p* < 0.001) in 2007 to 0.5852 (*p* < 0.001) in 2015, showing that inequalities slightly decreased over the two time periods, the difference being −0.0472 (*p* < 0.001). In 2007, the inequality in skilled birth attendance was mainly explained by the direct effect of prenatal care use (33.91%); living in a rural area (29.8%); economic status of household, namely being poorest (Q1) (21.4%) or poor (Q2) (4.13%); and the number of living children, namely four or more (16.88%). The rank of the key determinants slightly changed in 2015: being poorest (Q1) was the main determinant (23.5%), followed by living in rural area (20%), the prenatal care use came in the third position (15.7%), and then the number of living children being four or more (9.4%). The contribution of secondary education in inequality (by contrast with high education) showed a slight increase, from 2.7% in 2007 to 4.6% in 2015, while the contribution of women’s age to inequality remained insignificant in both surveys.

The component of corrected concentration in 2007 included in the model explains 112% of the overall socioeconomic inequality in skilled birth attendance. About −12% of the overall inequality is due to the residuals. In 2015, the determinants included in the model explained 85% of the overall socioeconomic inequality in SBA. About 15% of the overall inequality was due to the residuals. Residual variables contributed to some extent to the pro-rich inequality in SBA in 2015, whereas the sign of the residual variable’s contribution was negative in 2007. The unexplained part of inequality refers to the portion of the inequality in SBA that cannot be explained by systematic variation in the determinants across socioeconomic group, and consequently, it cannot be decomposed. Clearly, there must be other factors that account for this unexplained part of inequality, but data for those determinants were not collected.

### 3.4. Decomposing Changes in Inequality

Oaxaca-type decomposition measured the extent to which the changes in CIE contributions were due to changes in elasticities or to changes in inequality of the explanatory variables. Table 3 presents the results for two types of Oaxaca decomposition. The second and third columns showed the results based on Equation (7), whereas the fourth and fifth columns demonstrated results based on Equation (8). Regardless of the equation used, change in the decomposed concentration index between 2007 and 2015 is as shown in the final columns (sixth and seventh columns) in the table.

Each row in Table 3 is interpreted in terms of its contribution to the observed change in SBA in 2007 and 2015. A negative sign on a variable denotes that its contribution increases SBA inequalities over the period 2007–2015, while a positive sign indicates a reduction.

The results suggest that socioeconomic inequalities between women in the first, the second, and the fifth wealth quintile worsened SBA inequality over the period 2007–2015. Meanwhile, the improvement in SBA inequalities over that period resulted from reduced socioeconomic inequalities relating to rural residence and prenatal care use. We also found that the number of living children played a role in explaining SBA inequalities. Socioeconomic inequalities experienced by those with four or more children exerted a greater impact on SBA inequalities than any experienced by their counterparts with fewer children. One can infer that the number of living children (four or more) might affect maternal health-seeking behavior.

Similarly, over the same period, socioeconomic inequalities between those with secondary education—compared to those with higher education—contributed to SBA inequalities, as did socioeconomic inequalities among those residing in Hodh Charghy, Hodh Gharby, Assaba, Gorgol, and Guidimagha compared to those in Nouakchott.

Overall, changed CIE and elasticities of the determinants contributed in a different way to the reduction in inequalities in SBA. Prenatal care use (267.2%) accounted for the largest contribution to the observed decrease in inequality, due to changes in both elasticity and the unequal distribution. Rural location (151.1%) and having four or more living children (111.8%) also accounted for large contributions to the observed decrease in inequality, which is mainly due to changes of these two variables in elasticity rather than to the unequal distribution. In contrast, changing inequalities for wealth quintiles and primary education level appear to be more important than changing elasticities.

The change in residual variables shows the extent to which the residual variables contributed to the pro-rich inequality in SBA in 2015, whereas the sign of the residual variables’ contribution was negative in 2007.

## 4. Discussion

The study measured and explored changes in socioeconomic inequality in skilled birth attendance in Mauritania from 2007 to 2015. The main findings were as follows: (i) there was a pro-rich inequality in SBA use in both years; (ii) inequality in SBA decreased over time; (iii) the main contributors to inequality in SBA use were prenatal care use, living in a rural area, and economic status of household (being poor (Q2) or the poorest (Q1)); (iv) the rank of main contributors changed over that period; (v) improvement in prenatal care use and rural area characteristics; and (vi) having four or more children were accountable for the bulk of narrowing in SBA use inequality in Mauritania.

Inequality in SBA use in Mauritania did favor well-off women in 2007 and 2015. This finding is consistent with the results of several previous studies in Sub-Saharan Africa and low and middle-income countries [18,19,20,21,22,23,24,25,26,27,28,29,30,31,32]. Although pro-rich inequality persists, the decline by −0.047 points or −7.5% in inequality in SBA in Mauritania between 2007 and 2015 suggests a better performance than the average changes of inequality observed in Sub-Saharan Africa, namely +0.0311 points or +23.6% [22]. It also suggests a better performance than the average decrease in West Africa, namely −0.0205 points or −6.8%. However, the decline in inequalities in SBA in Mauritania was lower than the finding from Senegal −0.0916 points or −32.2% [22].

Between 2007 and 2015, several interventions were carried out by the government to improve the availability and deployment of qualified health staff, such as the creation of four public health schools in rural areas to make human resources available and maintain them in these areas. Annual recruitment and deployment of human resources within the country have accelerated. Furthermore, the increase in personnel has been accompanied by a doubling over the period of obstetric care facilities, and their concomitant modernization. These efforts would have contributed to the observed decrease in inequality in maternal health care services, particularly prenatal care, as well as the observed inequalities between rural and urban areas. Despite these efforts, the country remains far from the minimum threshold of 34.5 doctors, nurses, and midwives per 10,000 population that was established by WHO as being necessary to deliver essential maternal and child health services [49]. Data from National Health Statistics Yearbook show an increase in the number of qualified health personnel per 10,000 inhabitants at the rural level from 2.8 per 10,000 population in 2007 to 4.7 per 10,000 inhabitants in 2015, against 5.6 and 6.3 per 10,000 inhabitants for the urban area between 2007 and 2015, respectively [50,51].

The decomposition approach showed that the largest contributors to inequality in SBA use were prenatal care use and living in a rural area in 2007, and economic status of household women in 2015. This finding suggests that the change in SBA inequality is most sensitive to these three important determinants. Previous studies showed a significant and similar effect of prenatal consultations on skilled birth attendance [52]. Reduction of prenatal care contribution to inequality (by 18.81 percent) over those periods can be due to comprehensive policies targeting maternal care coverage as the first priority of the National Health Development Plan 2012–2015. In addition, the decline in rural–urban inequality explains, to some extent, the reduction in SBA inequality between 2007 and 2015. The economic rural–urban disparity triggered many challenges in the field of health care, such as geographic and financial access to health services [53,54,55,56,57]. The rural areas in Mauritania are characterized by extremely dispersed settlement patterns and very low densities, largely the result of water scarcity, small overall populations, huge distances to health facilities, and extremely low service standards. However, the recent developments in terms of rehabilitation and extension of the road network—where the combined length of paved roads increased from 1760 km in 2001, to 3069 km in 2010 before reaching 4867 km in 2014—might have helped to address transportation barriers to health care access particularly for rural and remote areas. In addition, in terms of health infrastructure, the number of health posts and health centers experienced an annual increase of 6% and 10% between 2008 and 2015, respectively. Consequently, geographic access to health care facility within 5 km increased from 58.7% in 2008 to 82.2% in 2015. Nevertheless, the rural service delivery challenges persisted for the poorest in remote areas. Despite the overall serious shortage of health workers, which limits the delivery and uptake of reproductive, maternal, neonatal, child health, and nutrition (RMNCHN) services, government resources allocated to rural areas have helped, to some extent, to mitigate urban–rural disparities in the distribution of RMNCHN care and skilled birth attendants.

In addition, the results of this study are in line with findings from low- and middle-income countries, suggesting that wealth is a key contributor to inequalities in skilled birth attendance. During 2007–2015, the country’s economic situation improved and the poverty rate dropped significantly. The country has achieved large but volatile growth in recent years and has recently been classified as a lower middle-income country (LMIC), with a GDP per capita of US$ 1524 in 2015 [3]. The prevalence of poverty has fallen from 45 $ in 2008 to 31% in 2014 [1,2]. However, women in the poorest and poor quintiles groups have been left behind as their contribution to inequalities in SBA increased by 3.6% over the period 2007–2015.

Several limitations of the study should be considered. Firstly, MICS surveys do not collect data on household expenditures or income. Instead, they measure the economic situation of households by analyzing household assets and housing characteristics, and create an assets-based Wealth Index. Secondly, distances between households and health facilities were not captured by the MICS survey (that would have allowed us to measure geographic accessibility to maternal health care services and include this variable in the model). Third, the only data available on financial risk protection were data on the voluntary insurance scheme covering maternal health costs in public health facilities in 2015 only. Data on this insurance scheme were not captured in the previous surveys. We measured its contribution to inequalities in 2015 and this seems to be limited to 5%. In the absence of data on this insurance scheme in surveys prior to 2015, we were not able to measure the contribution of this variable to changes in inequality over time; hence its exclusion from the analysis. Finally, the frequency of prenatal visits was not captured in the 2007 survey; consequently, we were unable to use the fourth prenatal visit as a variable for prenatal care.

## 5. Conclusions

Four key messages emerge from the analysis. First, large inequalities persist in childbirth assisted by a health professional. Although all women should receive skilled care during pregnancy and at delivery—skilled birth attendance at every delivery significantly reduces the risk of maternal and neonatal mortality—poor women and women living in rural areas still tend to use SBA services to a lesser extent than wealthy and urban women do. The concentration curves diverge from the line of equality and the concentration indices are significantly different from zero, showing a pro-rich distribution for both surveys covered by the research. Second, in both surveys, the main contributors to the pro-rich inequality in SBA were non-need factors, such as wealth index, prenatal care, and urban rural residency. This pattern does not accord with equity of access to SBA services, namely that women with the same health needs should have similar access to health care services.

Third, time-trends revealed a rather modest narrowing of the gap between the richest and the poorest. In 2007, prenatal care and the place of residence of women widely affects maternal health care use, while the Wealth Index influences maternal health care use to an important but lesser extent. In 2015, the order of importance changed, and it is the Wealth Index that becomes the primary factor, followed by the place of residence and the use of prenatal care. The number of living children, the level of education and age of women influence skilled birth attendance use to an important but lesser extent than Wealth Index, place of residence or prenatal care use. Finally, Oaxaca decomposition analysis suggests that the decrease in pro-rich inequalities in SBA is mainly due to the decreased disparities between urban and rural areas and prenatal care.

The results of this study provide a basis for several policy recommendations. First, removing financial barriers evident in the persistent link between access to skilled birth attendance and poverty will require augmenting existing programs such as the ongoing results-based financing project and the obstetric risk insurance scheme. The results-based financing is providing financial incentives to improve access to health care including maternal health care for poor families in rural regions through the indigence fund. Overall, removing user fees and improving financial protection will improve financial access to SBA services. Second, women using prenatal care tended to be more likely to have deliveries assisted by health professionals; this suggests a focus on improving the uptake of prenatal care. Third, regional status and urban−rural residence were found to be significant predictors of deliveries assisted by health professionals indicating that government’s reproductive and maternal programs should be intensified in the regions of Hodh Chargui, Hodh Ghabri, Assaba, Gorgol, and Guidimagha in order to further reduce the inequalities in the use of maternal health care use. Fourth, level of education was found to have an important impact on skilled birth attendance, suggesting that improving women’s educational level may have an impact on improving the use of skilled birth attendance in the future.

Other areas of intervention include communication for behavioral change and taking account of any cultural factors that impede uptake of reproductive health services.

This research found that inequalities are still high. Further research is needed to measure the burden of out-of-pocket payments on affordability of maternal health care including catastrophic health payment and impoverishing health payment.

## Figures and Tables

**Figure 1 ijerph-19-03566-f001:**
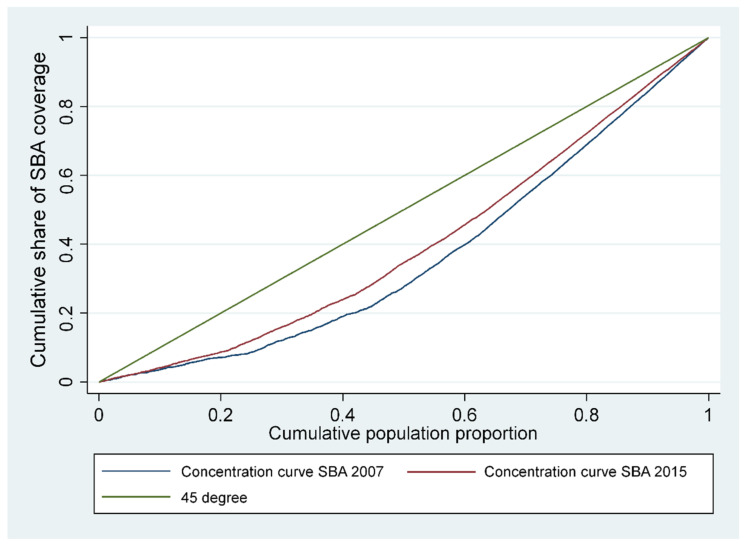
Concentration curves of SBA (2007–2015).

**Table 1 ijerph-19-03566-t001:** Characteristics of the sample.

	2007			2015		
	*n* = 3539	%	SBA Weighted Mean	*n* = 4172	%	SBA Weighted Mean
Age						
15–24	1167	33%	63%	1195	28%	66%
25–34	1583	45%	55%	1958	47%	65%
35–44	736	21%	54%	946	23%	62%
45–49	53	1%	31%	73	2%	50%
Birth order						
0	38	1%	80%	34	1%	68%
1	644	18%	70%	789	19%	77%
2	633	18%	65%	731	18%	70%
3	569	16%	56%	682	16%	69%
4	1655	47%	49%	1936	46%	56%
Prenatal						
yes	2860	81%	66%	3641	87%	70%
no	679	19%	18%	531	13%	27%
Place of residence						
urban	1509	43%	86%	1877	45%	87%
rural	2030	57%	36%	2 295	55%	46%
Region						
Hodh Charghy	423	12%	34%	502	12%	40%
Hodh Gharby	340	10%	27%	412	10%	38%
Assaba	371	10%	35%	479	11%	55%
Gorgol	341	10%	32%	508	12%	49%
Brakna	331	9%	58%	428	10%	82%
Trarza	333	9%	80%	299	7%	82%
Adrar	61	2%	64%	20	0%	66%
Nouadibou	119	3%	86%	137	3%	96%
Tagant	76	2%	39%	23	1%	55%
Guidimagha	177	5%	25%	329	8%	25%
Tirs-Zemour	63	2%	88%	20	1%	89%
Inchiri	8	1%	85%	5	1%	81%
Nouakchott	896	25%	90%	1010	24%	95%
Education					0%	
No	1042	29%	40%	1055	25%	50%
Primary	1992	56%	58%	2371	57%	61%
Secondary	451	13%	88%	655	16%	89%
University	54	2%	89%	91	2%	90%
Wealth index						
Q1 (Poorest)	857	24%	19%	879	21%	25%
Q2	694	20%	36%	875	21%	48%
Q3	657	19%	67%	829	20%	71%
Q4	681	19%	86%	838	20%	90%
Q5	650	18%	90%	751	18%	94%
SBA	3539		57.12%	4172		64.41%

**Table 2 ijerph-19-03566-t002:** Decomposition of inequality in access to SBA in Mauritania and total change (2007–2015).

	2007					2015					
	*n* = 3539					*n* = 4172					
Variables	Marginal Effects dy/dx	Elasticity	CIE	Absolute Contribution	Percentage Contribution (%)	Marginal Effects dy/dx	Elasticity	CIE	Absolute Contribution	Percentage Contribution (%)	Change
Age											
15–24	Ref.					Ref.					
25–34	−0.0249	−0.0445	0.0167	−0.0007	−0.12%	0.0281	0.0527	0.0718	0.0038	0.6%	0.0045
35–44	0.0495	0.0412	−0.0437	−0.0018	−0.28%	**0.0712 ****	0.0646	−0.0398	−0.0026	−0.4%	−0.0008
45–49	−0.0663	−0.0040	−0.0200	0.0001	0.01%	**0.1136 ***	0.0080	−0.0212	−0.0002	0.0%	−0.0002
Birth order											
Child (0)	Ref.					Ref.					
Child (1)	**−0.1908 ***	−0.1392	0.1054	−0.0147	−2.32%	−0.0226	−0.0171	0.1077	−0.0018	−0.3%	0.0128
Child (2)	**−0.2391 ****	−0.1713	0.0832	−0.0143	−2.25%	−0.0712	−0.0499	0.0650	−0.0032	−0.6%	0.0110
Child (3)	**−0.2661 ****	−0.1714	0.0097	−0.0017	−0.26%	−0.0801	−0.0524	0.0624	−0.0033	−0.6%	−0.0016
Child (4+)	**−0.2807 ****	−0.5251	−0.2033	0.1068	16.88%	−0.1251	−0.2321	−0.2323	0.0539	9.2%	−0.0528
Prenatal											
yes	**0.2052 *****	0.6632	0.3234	0.2145	33.91%	**0.1386 *****	0.4837	0.1823	0.0882	15.1%	−0.1263
no	Ref.					Ref.					
Place of residence											
Rural	**−0.1001 *****	−0.2296	−0.8207	0.1884	29.80%	**−0.0626 ****	−0.1378	−0.8494	0.1170	20.0%	−0.0714
Urban	Ref.					Ref.					
Region											
Hodh Charghy	**−0.1506 *****	−0.0719	−0.1999	0.0144	2.27%	**−0.1947 *****	−0.0937	−0.2172	0.0204	3.5%	0.0060
Hodh Gharby	**−0.1715 *****	−0.0660	−0.1580	0.0104	1.65%	**−0.1947 *****	−0.0768	−0.1518	0.0117	2.0%	0.0012
Assaba	**−0.1548 *****	−0.0650	−0.1616	0.0105	1.66%	**−0.1221 *****	−0.0561	−0.1186	0.0067	1.1%	−0.0038
Gorgol	**−0.1552 *****	−0.0598	−0.1343	0.0080	1.27%	**−0.1951 *****	−0.0951	−0.1281	0.0122	2.1%	0.0042
Brakna	−0.0681	−0.0254	−0.0537	0.0014	0.22%	0.0235	0.0096	−0.0355	−0.0003	−0.1%	−0.0017
Trarza	0.0196	0.0074	0.0697	0.0005	0.08%	−0.0634	−0.0182	0.0506	−0.0009	−0.2%	−0.0014
Adrar	−0.0153	−0.0011	0.0009	0.0000	0.00%	−0.1386	−0.0026	0.0016	0.0000	0.0%	0.0000
Nouadibou	−0.0784	−0.0105	0.0883	−0.0009	−0.15%	0.0681	0.0089	0.0947	0.0008	0.1%	0.0018
Tagant	−0.0893	−0.0077	−0.0216	0.0002	0.03%	**−0.1564 ***	−0.0034	−0.0022	0.0000	0.0%	−0.0002
Guidimagha	**−0.2419 *****	−0.0484	−0.0664	0.0032	0.51%	**−0.3264 *****	−0.1028	−0.1114	0.0114	2.0%	0.0082
Tirs-Zemour	−0.0140	−0.0010	0.0392	0.0000	−0.01%	−0.0978	−0.0019	0.0140	0.0000	0.0%	0.0000
Inchiri	0.0009	0.0000	0.0033	0.0000	0.00%	−0.1298	−0.0006	0.0027	0.0000	0.0%	0.0000
Nouakchott	Ref.					Ref.					
Education Level											
Primary	0.0258	0.0467	0.0466	0.0022	0.34%	**0.0479 ***	0.0928	−0.1097	−0.0102	−1.7%	−0.0123
Secondary	**0.0704 ****	0.0508	0.3340	0.0170	2.68%	**0.0831 *****	0.0698	0.3569	0.0249	4.3%	0.0079
High	Ref.					Ref.					
Wealth Quintile											
Q1 (Poorest)	**−0.1888 *****	−0.1830	−0.7342	0.1343	21.24%	**−0.2423 *****	−0.2043	−0.6655	0.1360	23.2%	0.0016
Q2	**−0.1328 *****	−0.1042	−0.2506	0.0261	4.13%	**−0.1308 *****	−0.1097	−0.3092	0.0339	5.8%	0.0078
Q3	−0.0126	−0.0094	0.0464	−0.0004	−0.07%	−0.0355	−0.0282	0.0314	−0.0009	−0.2%	−0.0005
Q4	0.0293	0.0225	0.3392	0.0076	1.21%	0.0092	0.0074	0.3530	0.0026	0.4%	−0.0050
Q5 (Richest)	Ref.					Ref.					
Total observed				0.711	112%				0.5000	85%	
Residual				−0.0786	−12%				0.0851	15%	
CIE				0.6324	100%				0.5852	100%	

Statistically significant marginal effects are in bold (* *p* < 0.1, ** *p* < 0.05, *** *p* < 0.01).

**Table 3 ijerph-19-03566-t003:** Oaxaca-type decomposition for change in inequality (2007–2015).

	Equation (7)		Equation (8)			
	Weighted Δ in Elasticity	Weighted Δ in CIE	Weighted Δ in CIE	Weighted Δ in Elasticity	Total Δ	%Δ
Age						
15–24	Ref.					
25–34	0.0029	0.0016	−0.0025	0.0070	0.0045	−9.6%
35–44	0.0003	−0.0010	0.0002	−0.0009	−0.0008	1.6%
45–49	0.0000	−0.0002	0.0000	−0.0003	−0.0002	0.5%
Birth order						
Child (0)	Ref.					
Child (1)	0.0000	0.0129	−0.0003	0.0132	0.0128	−27.2%
Child (2)	0.0009	0.0101	0.0031	0.0079	0.0110	−23.3%
Child (3)	−0.0028	0.0012	−0.0090	0.0074	−0.0016	3.4%
Child (4+)	0.0067	−0.0596	0.0152	−0.0681	−0.0528	111.8%
Prenatal						
yes	−0.0682	−0.0580	−0.0936	−0.0327	−0.1263	267.2%
no	Ref.					
Place of residence						
Rural	0.0040	−0.0753	0.0066	−0.0780	−0.0714	151.1%
Urban	Ref.					
Region						
Hodh Charghy	0.0016	0.0044	0.0012	0.0047	0.0060	−12.6%
Hodh Gharby	−0.0005	0.0017	−0.0004	0.0016	0.0012	−2.6%
Assaba	−0.0024	−0.0014	−0.0028	−0.0010	−0.0038	8.1%
Gorgol	−0.0006	0.0047	−0.0004	0.0045	0.0042	−8.8%
Brakna	0.0002	−0.0019	−0.0005	−0.0012	−0.0017	3.6%
Trarza	0.0003	−0.0018	−0.0001	−0.0013	−0.0014	3.0%
Adrar	0.0000	0.0000	0.0000	0.0000	0.0000	0.0%
Nouadibou	0.0001	0.0017	−0.0001	0.0018	0.0018	−3.8%
Tagant	−0.0001	−0.0001	−0.0001	0.0000	−0.0002	0.3%
Guidimagha	0.0046	0.0036	0.0022	0.0061	0.0082	−17.4%
Tirs−Zemour	0.0000	0.0000	0.0000	0.0000	0.0000	0.0%
Inchiri	0.0000	0.0000	0.0000	0.0000	0.0000	0.0%
Nouakchott	Ref.					
Education Level						
Primary	−0.0145	0.0021	−0.0073	−0.0051	−0.0123	26.1%
Secondary	0.0016	0.0063	0.0012	0.0068	0.0079	−16.8%
High	Ref.					
Wealth Quintile						
Q1 (Poorest)	−0.0140	0.0157	−0.0126	0.0142	0.0016	−3.5%
Q2	0.0064	0.0014	0.0061	0.0017	0.0078	−16.5%
Q3	0.0004	−0.0009	0.0001	−0.0006	−0.0005	1.0%
Q4	0.0001	−0.0051	0.0003	−0.0053	−0.0050	10.6%
Q5 (Richest)	Ref.					
Total observed	−0.0730	−0.1380	−0.0934	−0.1176	−0.2110	446%
Residual					0.1637	−346%
CIE					−0.0473	100%

## Data Availability

Data described in this article are openly available at the UNICEF Multiple Indicator Cluster Surveys program website: https://mics.unicef.org/surveys (accessed on 30 August 2021).

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
