# Peer review of "Changes in Inequality in Use of Maternal Health Care Services: Evidence from Skilled Birth Attendance in Mauritania for the Period 2007–2015"

_ijerph, 2022, doi:10.3390/ijerph19063566_

Round 1

Reviewer 1 Report

Changes in Inequality in Utilization of Maternal Health Care Services: Evidence from Skilled Birth Attendance in Mauritania for the period 2007-2015. 

Reviewed for IJERPH

This manuscript engages with the literature on global public health and the social determinants of health to study the changes in skilled birth attendance (SBA) in Mauritania during a period of economic growth within the country.  SBA is method to make childbirth a safer process, thereby (hopefully) reducing both maternal mortality and infant mortality.  In this particular study, the authors make use of a concentration curve, concentration index, decomposition of the concentration index and a Oaxaca-type decomposition to study the socioeconomic variables that influence whether or not a skilled birth attendant was present at a childbirth.  Data was collected and analyzed for both the years 2007 and 2015. 

In Mauritania, the country of interest here, the authors found clear pro-rich inequality in SBA usage in both years surveyed.  They also found that this inequality decreased over time, and that the key drivers of this inequality was socioeconomic variables such as prenatal care distribution, rural residence, and the economic resources of the household in question.   The main reason for the narrowing of this inequality in SBA usage was improvements in prenatal care, rural area changes and the number of children. 

Overall, I find this to be a well cited, well researched and well explained study.  The data is solid and the methods applied to the questions make sense here.  It is well written and I am able to see how the data informs the conclusions reached by the authors.  I have no concerns with the data analysis.  For overall improvement, my only real suggestion would be to engage a bit with the literature regarding overcoming socioeconomic barriers to health in the discussion and conclusion sections.  For example, the reader is told that Mauritania’s gains are greater than the Sub-Saharan Africa average, but lag behind Senegal.  This leads me to wonder – what did Senegal do that put them at the top?  While the paper is obviously about Mauritania and not Senegal, I do think it would benefit from a short discussion as to what Mauritania can do going forward, as that doesn’t come across as well on the back end.  If the majority of SBA utilization is explained by socioeconomic variables, then we can address these socioeconomic variables through social policy. 

Otherwise, I commend the researchers for a well-researched, well written manuscript! 

Author Response

Dear reviewer,

Thank you very much for the comments and suggestions. Below some response to the comments below.

Kind regards

Reviewers’ comments

Authors Responses

Overall, I find this to be a well cited, well researched and well explained study.  The data is solid and the methods applied to the questions make sense here.  It is well written, and I am able to see how the data informs the conclusions reached by the authors.  I have no concerns with the data analysis

Thank you

.  For overall improvement, my only real suggestion would be to engage a bit with the literature regarding overcoming socioeconomic barriers to health in the discussion and conclusion sections.  For example, the reader is told that Mauritania’s gains are greater than the Sub-Saharan Africa average, but lag behind Senegal.  This leads me to wonder – what did Senegal do that put them at the top?  While the paper is obviously about Mauritania and not Senegal, I do think it would benefit from a short discussion as to what Mauritania can do going forward, as that doesn’t come across as well on the back end.  If the majority of SBA utilization is explained by socioeconomic variables, then we can address these socioeconomic variables through social policy

Thank you for this comment.

This will require additional data for analysis.  However, this important point might be the subject for future investigation including other neighboring countries such as Morocco for example.

 Otherwise, I commend the researchers for a well-researched, well written manuscript! 

Thank you

Reviewer 2 Report

Overall a great piece of research. While this paper is well written paper and very detailed, there are a couple of formatting inconsistencies, minor typos or upper case inconsistencies such as the following:

“…financing scheme under an universal health coverage vision”

“...In order to fill theses knowledge gap, the…”

“…principles of the Primary health Care focus…”

There are also a number of other areas for improvement and consideration.

  1. The abstract clearly states the study aim as follows, but the aim is not detailed in the article itself:

“This study aimed to measure and analyze the inequality in the 12 use of skilled births attendance services in Mauritania”

  1. Ensuring that this level of clarity if accessible within the article would be useful, to help improve article accessibility.
  2. Figure 2. Decomposition of inequality in access to SBA in Mauritania and total change 2007–2015 appears to include a number of lines that appear not to have any data, or limit data that is not accessible in the figure. Ensuring the Figure is accessible and that the respective lines include information would be useful. Perhaps a panel figure would be useful so that the reader can understand what the key messages are in this instance.
  3. Given the level of detail throughout the article, the conclusion and recommendations appear to be very vague and high level, without any detail. For example, “Therefore, this study recommends policies that promote development focused on eliminating the gaps between rich and poor.” Could the authors provide some more pragmatic conclusion and recommendation that are of tangible benefit to communities given the analysis that was undertaken?
  4. Overall, a useful article with some consistent key messages. However, the article could be made more accessible, with some pragmatic conclusions and recommendations.

Author Response

Thank you very much for the comments and suggestions. We have modified the manuscript according to the comments below.

Reviewers’ comments

Authors Responses

Overall a great piece of research.

Thank you

While this paper is well written paper and very detailed, there are a couple of formatting inconsistencies, minor typos or upper-case inconsistencies such as the following:

“…financing scheme under an universal health coverage vision”

“...In order to fill theses knowledge gap, the…”

“…principles of the Primary health Care focus…”

Reformulated: “financial protection in line with the universal health coverage objectives”

Dropped.

This was dropped in the revised version

  1. The abstract clearly states the study aim as follows, but the aim is not detailed in the article itself:

“This study aimed to measure and analyze the inequality in the 12 use of skilled births attendance services in Mauritania”

2.         Ensuring that this level of clarity if accessible within the article would be useful, to help improve article accessibility.

Thanks. We’ve attempted to clarify this in the conclusion and recommendations sections

  1. Figure 2. Decomposition of inequality in access to SBA in Mauritania and total change 2007–2015 appears to include a number of lines that appear not to have any data, or limit data that is not accessible in the figure. Ensuring the Figure is accessible and that the respective lines include information would be useful. Perhaps a panel figure would be useful so that the reader can understand what the key messages are in this instance.

The data is there but the contribution percentages of some variables are close to zero, that is why it appears as missing data.

We have revised the figure to illustrate the key contributors to the change inequality

  1. Given the level of detail throughout the article, the conclusion and recommendations appear to be very vague and high level, without any detail. For example, “Therefore, this study recommends policies that promote development focused on eliminating the gaps between rich and poor.” Could the authors provide some more pragmatic conclusion and recommendation that are of tangible benefit to communities given the analysis that was undertaken?

The conclusion and recommendations have been revised to focus on more pragmatic conclusion and recommendation.

  1. Overall, a useful article with some consistent key messages. However, the article could be made more accessible, with some pragmatic conclusions and recommendations.

The conclusion and recommendations has been revised with focus on more pragmatic conclusion and recommendation.

Reviewer 3 Report

Thank you very much for sending me this manuscript. I believe this study is very important and timely with sound methods. I have a few comments for the authors.

  1. Why did you focus on skilled birth attendance? There are indicators such as timing of the first ANC use and frequency of ANC visit.

  1. Would you consider other variables that possibly contribute to socioeconomic inequalities such as marital status, religion, and ethnicity?

  1. I enjoy that the authors note the possible importance of communication/behavioural change in the Limitation section. There is a variable that captures exposure to family planning messages. Is it a possibly important variable to include in the model?

  1. Can you strengthen your discussion on policy implications? What kind of existing policies/laws/initiatives and how does this study improve them?

Author Response

Thank you very much for the comments and suggestions. We have modified the manuscript according to the comments below.

Reviewers’ comments

Authors Responses

Thank you very much for sending me this manuscript. I believe this study is very important and timely with sound methods.

Thank you

  1. Why did you focus on skilled birth attendance? There are indicators such as timing of the first ANC use and frequency of ANC visit.

MICS 2007 does not collect data on the frequency of antenatal visits. The four visits were introduced later in 2012. The lack of data on the frequency of antenatal visits did not allow us to include four antenatal care visits in the model.

  1. Would you consider other variables that possibly contribute to socioeconomic inequalities such as marital status, religion, and ethnicity?

Unfortunately, religion, ethnicity data are not collected in MICS survey.  Marital status was not included. Few previous studies have included this variable in the equity model for assisted skilled birth attendance. In addition, the collection of some reproductive health data in the MICS survey (Mauritania) was limited to married women only for cultural reasons.

  1. I enjoy that the authors note the possible importance of communication/behavioral change in the Limitation section. There is a variable that captures exposure to family planning messages. Is it a possibly important variable to include in the model?

Thank you.

The variables were chosen based on literature review and the possible correlation shown in previews studies. For that reason, we did not include exposure to family planning messages in the model. This variable might be added in future update of the study.

  1. Can you strengthen your discussion on policy implications? What kind of existing policies/laws/initiatives and how does this study improve them?

The conclusion and recommendations revised with focus on more pragmatic conclusion and recommendation based on existing policies/laws/initiatives